# Structural Characterization and Antioxidant Activity of Exopolysaccharide from Soybean Whey Fermented by *Lacticaseibacillus plantarum* 70810

**DOI:** 10.3390/foods10112780

**Published:** 2021-11-12

**Authors:** Juanjuan Tian, Qingyan Mao, Mingsheng Dong, Xiaomeng Wang, Xin Rui, Qiuqin Zhang, Xiaohong Chen, Wei Li

**Affiliations:** College of Food Science and Technology, Nanjing Agricultural University, Nanjing 210095, China; 2018208012@njau.edu.cn (J.T.); qingyanmao@njau.edu.cn (Q.M.); dongms@njau.edu.cn (M.D.); 2019208028@njau.edu.cn (X.W.); ruix@njau.edu.cn (X.R.); zqq@njau.edu.cn (Q.Z.); xhchen@njau.edu.cn (X.C.)

**Keywords:** *Lacticaseibacillus plantarum* 70810, fermentation, soybean whey, polysaccharide, structure characterization, antioxidant activity

## Abstract

Soybean whey is a high-yield but low-utilization agricultural by-product in China. In this study, soybean whey was used as a substrate of fermentation by *Lacticaseibacillus plantarum* 70810 strains. An exopolysaccharide (LPEPS-1) was isolated from soybean whey fermentation by *L. plantarum* 70810 and purified by ion-exchange chromatography. Its preliminary structural characteristics and antioxidant activity were investigated. Results show that LPEPS-1 was composed of mannose, glucose, and galactose with molar ratios of 1.49:1.67:1.00. The chemical structure of LPEPS-1 consisted of →4)-α-D-Glc*p*-(1→, →3)-α-D-Gal*p*-(1→ and →2)-α-D-Man*p*-(1→. Scanning electron microscopy (SEM) revealed that LPEPS-1 had a relatively rough surface. In addition, LPPES-1 exhibited strong scavenging activity against DPPH and superoxide radicals and chelating ability on ferrous ion. This study demonstrated that soybean whey was a feasible fermentation substrate for the production of polysaccharide from *L. plantarum* 70810 and that the polysaccharide could be used as a promising ingredient for health-beneficial functional foods.

## 1. Introduction

*Lacticaseibacillus* are the largest genus of lactic acid bacteria (LAB), and more than 50 different species of LAB belong to the genus *Lacticaseibacillus*. The *Lacticaseibacillus* mainly exist in the human gut and traditional fermented foods [1]. People have isolated different *Lacticaseibacillus plantarum* from Kimchi, fermented fish, and fermented drinks [2,3,4]. Study has shown that *L. plantarum* are safe and can produce by fermentation a variety of secondary metabolites that are beneficial to human health, such as lactic acid, acetic acid, exopolysaccharide, specific proteases, and bacteriocins [5]. Polysaccharide is biopolymer, and it is widely distributed in all animals, plants, fungi, and bacteria. Bacterial polysaccharides, especially those exopolysaccharide (EPS) produced by LAB, have been widely used in the food industry [6]. EPS from LAB has been reported to exhibit several types of biological activity, such as immunomodulatory, antioxidant, and antitumor activity [7,8,9]. It has been reported that the activity of EPS is related to its structural characteristics, such as its monosaccharide composition, molecular weight, types of glycosidic linkages, as well as its chain conformation [10]. Hence, research on the relationship between structure and activity on polysaccharide has become a hot topic.

Soybean products are a traditional food in China. Soybean whey is a by-product that is generated from the preparation of soybean products [11]. For a long time, soybean whey was considered to be a waste product by the food industry, and most soybean whey is directly discharged into the environment without treatment, causing environmental pollution [12]. In fact, soybean whey is a rich source of proteins, carbohydrates, oligosaccharides, and some trace elements (calcium, iron, magnesium, etc.), and it should not be classified as waste [13,14]. Some experts are developing new ideas on how to turn soybean whey into high value products. Researchers found that soybean whey is rich in nutrients and that it could be used as a substitute for microbial culture medium. Tu et al. [15] used water kefir grains to ferment soybean whey, which can significantly increase the content of flavonoids, phenolics, and isoflavones and improve antioxidant capacity. Others have reported that soybean whey could be converted into a soybean alcoholic beverage by fermentation of yeast [16]. Microbial fermentation is a useful biotechnological strategy that can turn soybean whey into bioactive ingredients. In the present research, *L. plantarumn* 70810 was isolated from traditional Chinese Sichuan pickles by our laboratory. In our previous study, polysaccharide from *L. plantarumn* 70810 fermentation by a synthetic liquid medium was studied, and the polysaccharide exhibited potent antioxidant and antitumor activities [17,18]. However, polysaccharide from *L. plantarumn* 70810 fermented soybean whey has not attracted attention. It was reported that the production of microbial EPS was affected by the composition of the culture medium, growth conditions, the age of the strain, and the amount of inoculation [19]. Among them, the composition of the medium, especially the carbon source and nitrogen source, had a greater impact on the growth of microbial compounds and the production of EPS [19]. Therefore, it is meaningful to study the production of polysaccharide by *L. plantarum* 70810 in soybean whey medium.

Therefore, in this study, the EPS from soybean whey fermented by *L. plantarum* 70810 was prepared and characterized by ultraviolet (UV) spectrophotometry, Fourier transform infrared (FI-IR) spectrophotometry, gas chromatography (GC), scanning electron microscopy (SEM), and nuclear magnetic resonance imaging (NMR). In addition, the in vitro antioxidant activity of EPS against DPPH and superoxide anion and the chelating activity on ferrous ion were evaluated.

## 2. Materials and Methods

### 2.1. Materials and Chemicals

The strain *L. plantarum* 70810 was isolated in our laboratory from traditional Chinese Sichuan pickles [20]. The strain was activated by De Man Rogosa Sharpe (MRS) liquid medium, which (1 L) contained 20 g of glucose, 10 g of beef cream, 10 g of peptone, 5 g of anhydrous sodium acetate, 5 g of yeast extract, 2.62 g of potassium dipotassium phosphate, 2 g of triammonium citrate, 0.58 g of magnesium sulfate heptahydrate, 0.198 g of manganese sulfate, and 1.0 mL of Tween 80. The soybean whey was obtained from Nanjing Fruit and Fruit Food Co., Ltd. The soybean whey contained a total solids content of 18.2 ± 1.3 g/L, including 4.5 g/L protein, 8.5 ± 0.4 g/L carbohydrate (predominantly stachyose and sucrose), and 3.4 ± 0.3 g/L total ash.

DEAE-cellulose-52 was purchased from Waterman Co. Ltd. (Springfield, United States,). Dialysis membrane with a molecular weight of 8000–10,000 Da was purchased from Solarbio Co., Ltd. (Beijing, China). D-galactose, flupromazine, 2,4,6-tripyridazine (TPTZ), nitrotetrazolium chloride (NBT), phenazine methyl sulfate (PMS), and reducing coenzyme I (NADH) were purchased from Sigma-Aldrich (St. Louis, MO, USA). All other reagents were analytical grade and were purchased from the Sinopharm Chemical Reagent Co., Ltd. (Shanghai, China).

### 2.2. Isolation and Purification of EPS

The isolation of EPS referred to the previous method [21]. After fermentation at 37 °C for 24 h in the soybean whey, the fermentation supernatant was obtained by centrifugation (4 °C, 12,000 rpm, 10 min). An amount of 80% (*w*/*v*) TCA was added to the supernatant, resulting in a final concentration of 4% TCA in the supernatant. This step was to remove the protein from the supernatant. Then, the solution was centrifuged, concentrated, and precipitated with 95% ethanol at 4 °C for 12 h. The precipitate was dissolved in the deionized water and dialyzed for 3 days in the running water. The crude EPS was obtained after lyophilization.

The crude EPS was further separated and purified by DEAE-52 cellulose column (2.6 × 30 cm). The polysaccharide solution was eluted with deionized water and 0.1 and 0.3 M NaCl at a flow rate of 1.0 mL/min. An amount of 10 mL of elution was collected in each tube, and the total sugar content was measured by phenol-sulfuric acid method [21]. The collected two fractions were named LPEPS-1 and LPEPS-2.

### 2.3. Analysis of Monosaccharide Composition

Monosaccharide composition was detected by GC according to our previous method [22]. First, the polysaccharide was hydrolyzed with trifluoroacetic acid (TFA). An amount of 5 mg of sample was hydrolyzed with 2 mL TFA (2 M) in oil bath (120 °C) for 2 h. The hydrolyzed solution was evaporated to dryness under reduced pressure. Then, the hydrolysates were derivatized with aldononitrile acetate. GC analysis was on an Agilent 6890N GC equipped with a flame ionization detector (FID). An HP-5 capillary column (30 m × 0.32 mm i.d., 0.25 μm) was used. The column temperature program was as follows: the initial temperature was 120 °C and maintained for 3 min, and then the temperature was increased to 210 °C for 4 min at a rate of 15 °C/min. The temperature of the injector and detector were 250 °C and 280 °C, respectively. The nitrogen flow rate was 1 mL/min. The composition and molar ratio of monosaccharides in the sample were determined by comparing the retention time and chromatographic peak area of the sample and the standard sample

### 2.4. UV and FT-IR Spectra Analysis

An amount of 1.0 mg/mL sample was prepared for UV (U-4100, Hitachi Ltd., Tokyo, Japan) measurement in a wavelength range of 190–500 nm. An amount of 1.0 mg of dried sample was mixed with 120.0 mg KBr and then pressed into pellet for FT-IR spectrum analysis (Bruker Co, Ettlingen, Germany) from 500 to 4000 cm^−1^.

### 2.5. Nuclear Magnetic Resonance (NMR) Analysis

An amount of 60 mg of the sample was dissolved in 1.0 mL D_2_O (99.9% D, Aladdin Co., Ltd. Shanghai, China). The one-dimensional (1D) (^1^H, ^13^C) and two-dimensional (2D) (^1^H-^1^H COSY, ^1^H-^13^C HSQC) NMR experiments were performed on a Bruker AVANCE AV-500 spectrometer (Bruker Group, Fällanden, Switzerland). The residual solvent (D_2_O) signal was used as an internal standard. The chemical shifts (*δ*) were described in parts per million (ppm).

### 2.6. Scanning Electron Microscopy (SEM) Analysis

The surface structural characteristics of the sample were observed using an S-3000 scanning electron microscope (FESEM, S-4800, Hitachi, Japan). The dried sample was sputter-coated with a gold layer before observation, and then the images were recorded by using the SEM.

### 2.7. Particle Size and Zeta-Potential

The EPS solution (0.2 mg/mL) was used for zeta-potential and hydrodynamic diameter (Z-average) determination. The experiments were carried out using a Malvern Zeta-size (Malvern Instruments, UK) with the measurement temperature at 25 °C.

### 2.8. Antioxidant Activities In Vitro

#### 2.8.1. Scavenging Activity of DPPH Radicals

DPPH free radical scavenging activity was determined by referring to the Qiao et al. [23] method with slight modifications. Different concentrations (0.125, 0.25, 0.5, 1, 2, and 4 mg/mL) of LPEPS-1 were prepared. The 0.4 Mm DPPH in ethanol was prepared. Amounts of 1.0 mL of the polysaccharide solution, 0.2 mL DPPH solution, and 1.8 mL deionized water were mixed and then incubated for 30 min at room temperature. The absorbance at 517 nm was measured by using a spectrophotometer. Ascorbic acid (Vc) was used as positive control. The scavenging activity of the DPPH radical was calculated by the following equation:Scavenging activity (%) = A _blank_ − A _sample_/A _blank_ × 100(1)

#### 2.8.2. Scavenging Activity of Superoxide Anion Radical

The scavenging activity of the superoxide anion radical was carried out by the method described by Qi et al. [24]. The reaction mechanism is that PMS reacts with NADH to produce superoxide anions. The generated superoxide anion can react in color with NBT. The colored substance has a maximum absorption wavelength at 560 nm. If the polysaccharide solution has the ability to clear superoxide anions, its reaction color will become lighter. The lower the absorbance, the stronger the removal ability of superoxide anions. Briefly, 72 μM NBT, 30 μM PMS, and 338 μM NADH were prepared with 0.1 M phosphoric acid buffer (pH 7.4). Different concentrations (0.125, 0.25, 0.5, 1, 2, and 4 mg/mL) of LPEPS-1 were prepared. The sample solution was added to NBT, PMS, and NADH at ratios of 1:1:1:1, and the mixture solution was incubated for 5 min at room temperature. The absorbance was measured at 560 nm. The scavenging activity on superoxide anion radical was calculated as follows:Scavenging activity on superoxide anion radical (%) = A_blank_ − A_sample/_A _blank_ × 100(2)

#### 2.8.3. Chelating Metal Ion Capacity

The ability of the chelating metal ion was determined by Liu et al. [25]. Fecl_2_ can chelate with Felozine to form Felozine-Fe^2+^; however, when polysaccharide solution is added, the polysaccharide can chelate with Fe^2+^, thereby inhibiting Felozine–Fe^2+^ formation. The chelating rate was calculated by measuring with a UV spectrophotometer the absorbance change before and after adding the polysaccharide. Different concentrations (0.125, 0.25, 0.5, 1, 2, and 4 mg/mL) of LPEPS-1 were prepared. The 1.0 mL sample, 0.05 mL FeCl_2_ (2 mM) and 0.2 mL ferrozine (0.2 mM) were mixed together. Total volume was up to 4 mL with distilled water. Then, the mixture solution was shaken evenly and reacted for 10 min at room temperature. Absorbance was measured at 562 nm. The chelating ability of the ferrous ion is given bellow:Metal chelating effect (%) =A _blank_ − A _sample_/A _sample_ × 100(3)

### 2.9. Statistical Analysis

All the experiments were carried out in triplicate. The results are expressed as means ± standard deviation (SD). The experiment’s data analysis was conducted using SPSS 23.0 software and Origin 2018 software.

## 3. Results and Discussion

### 3.1. Isolation and Purification of EPS

The crude EPS was isolated and purified from the soybean whey fermented by *L. plantarum* 70810. In order to obtain the purified EPS fraction, DEAE-52 cellulose column was used to separate and purify polysaccharide. As shown in Figure 1a, two fractions, LPEPS-1 and LPEPS-2, were obtained with the yield of 78.7% and 21.3%, respectively. Considering that the yield of LPEPS-2 was relatively low, we therefore mainly studied the structure and antioxidant activity of LPEPS-1.

### 3.2. UV and FT-IR Spectra Analysis

The characteristic UV spectrum of LPEPS-1 was observed from 190 nm to 500 nm. Figure 1b shows that LPEPS-1 did not contain nucleic acid and protein because it had no absorption peaks at 260 nm or 280 nm. In the wavelength range of 4000–500 cm^−1^ (Figure 1c), LPEPS-1 exhibited typical absorption peaks of polysaccharide. The wide and strong absorption peak at 3395 cm^−1^ was the stretching vibration of the hydroxyl group (-OH) [26]. The strong absorption peak at 2936cm^−1^ was the stretching vibration of the C-H bond group. The signal strong absorption peak at 1632 cm^−1^ was related to the asymmetric stretching vibration of the C=O bond. The weak absorption peak at 1437 cm^−1^ was the angular vibration of the C-H bond [27]. The wave numbers between 1200 and 1000 cm^−1^ were dominated by the glycosidic linkage C-O-H and C–O–C stretching vibration contribution [28].

### 3.3. Monosaccharide Composition of LPEPS-1

The monosaccharide composition analysis result of LPEPS-1 obtained by acid hydrolysis is shown in Figure 1d. Comparing the retention time with standard sugars, the results indicate that LPEPS-1 was composed of mannose, glucose, and galactose, with molar ratios of 1.49:1.67:1.00. This monosaccharide composition result was consistent with that reported by Wang et al. [17]. Meanwhile, Wang et al. [18] cultured *L. plantarum* 70810 in a semi-defined medium, and the EPS from *L. plantarum* 70810 only contained glucose. Ismail et al. [29] reported that the EPS from *L. plantarum* was composed of glucose and mannose, with a molar ratio of approximately 2:1. These different results indicate that strains, culture conditions, and medium composition affect the production of EPS by *L. plantarum*.

### 3.4. NMR Characterization of EPS

The anomeric region signals of *δ* 4.5–5.5 ppm were analyzed in the ^1^H NMR spectrum. Normally, there are three proton signals in this region, which means that three kinds of glycoside bonds exist. As shown in the Figure 2a, three proton signals occurred at *δ* 5.37, 5.32, 4.94 ppm, and they are labeled as A, B, and C, respectively. The ^13^C spectrum of LPEPS-1 was distributed in the range of *δ* 60–110 ppm (Figure 2b). The anomeric C-1 signals were detectable at *δ* 102.57, 102.19, and 101.10 ppm. According to the chemical shift (*δ*) and coupling constant value (*J*) of the anomeric proton, it could be judged that A, B, and C were all α-configurations. The complete spectra of ^1^H and ^13^C were assigned by 2D COSY and HSQC. From the ^1^H-^1^H COSY spectrum (Figure 2d), the chemical shifts of H-2 to H-6 were determined. These signals cross-linked to the proton signals at *δ* 5.37/3.58, 3.58/3.93, 3.93/3.67, 3.67/3.78, and 3.78/3.82. Taken together with the ^1^H-^13^C spectrum of LPEPS-1, the relevance between protons and carbons could be appointed. The signals of C-1 at δ102.57, 102.19, and 101.10 ppm were assigned to B, A, and C, respectively. The C2–C6 resonance signals of residue A were assigned to *δ* 73.88, 75.13, 79.52, 73.88, and 63.11 ppm, respectively. The C-4 signal of residue **A** had moved downfield to 79.52 ppm, which indicated that C-4 of residue A was substituted [30]. In addition, the DEPT-135 spectrum also showed chemical shifts with the ^13^C spectrum. However, the DEPT-135 spectrum exhibited positive CH_3_ and CH signals and negative CH_2_ signals. In Figure 2c, negative peaks in DEPT-135 showed a signal CH_2_ at *δ* 63.19 ppm, which was attributed to the C-6 of unsubstituted carbon of A, B, and C residues. When combined with the monosaccharide composition, NMR analysis, and literature data, residue A was identified as →4)-α-Glc*p* (1→ [31,32]. In the same way, residues B and C were identified as →2)-α-Man*p* (1→ and →3)-α-Gal*p* (1→, respectively [33,34,35,36]. The ^1^H NMR and ^13^C NMR chemical shifts of residues A–C are summarized in Table 1.

### 3.5. SEM Analysis

SEM is an important tool for characterizing the advanced structure of polysaccharide, which is a good technique for analyzing structural morphology, including size, shape, and porosity [37]. The morphology of a polysaccharide surface can be observed intuitively by magnification at different times. In Figure 3, the surface of LPEPS-1 displays a sheet-like appearance, and its surface is slightly rough with holes of different sizes, showing a compact and uneven distribution under the microscope. The structural micrographs were different from the SEM images of EPS from *L. plantarum* WLPL04 [30], which showed an irregular structure resembling sheets. In addition, EPS from *L. plantarum* KX041 presented a flake-shape structure with a rough surface [38]. The differences in microstructure and surface morphology of EPS may be related to the physicochemical properties, the selective type of extraction, the method of drying, and the structure of polysaccharide [39,40].

### 3.6. Particle Size and Zeta-Potential Analysis

Particle size is an important parameter of a substance in a solution. It can reflect the stability of the solution and decide the substance’s potential application prospects [41]. In different environments, polysaccharide always shows different conformations, which can be expressed by apparent particle size on a laser particle size analyzer. Han Q et al. [42] reported that tea (*Camellia sinensis*) flower polysaccharide solution could produce larger size particles by heat treatment, compared with those produced before heating. In this study, the particle size distribution of LPEPS-1 with concentration of 0.2 mg/mL in aqueous solution was determined by a laser particle size analyzer. The results are shown in Figure 4a,b. The mean diameter of LPEPS-1 was 34.22 nm. The size distribution of LPEPS-1 was narrow and uniform, indicating that LPEPS-1 was homogeneous polysaccharide. The zeta potential is also an important parameter that reflects the repulsive force between polysaccharide molecules in solution. In general, large repulsive force can prevent particles from aggregation [43]. LPEPS-1 was negatively charged, and the zeta potential was –2.93 ± 0.13 mv. Due to the charge polarities, the particles were not massively aggregated. Taken together, LPEPS-1 had good stability.

### 3.7. Analysis of Antioxidant Activity In Vitro

#### 3.7.1. DPPH Radical Scavenging Activity

DPPH free radical has become a free radical that is commonly used to evaluate the antioxidant activity of various polysaccharides. The antioxidant mechanism of DDPH free radicals is that it can accept hydrogen protons to become non-radical DPPH-H, which leads to the disappearance of DPPH purple. According to the change of purple color before and after adding polysaccharide samples, the DPPH scavenging rate can be calculated [44]. The antioxidant property of LPEPS-1 was evaluated with DPPH, which compared with that of Vc. In Figure 5a, no significant change can be seen in the scavenging rate of LPEPS-1 on DPPH, which did not change significantly at the selected concentrations range (0.125 mg/mL to 4.0 mg/mL). However, it can be seen that the scavenging ability of DPPH still had a dose-dependent relationship. The scavenging activity of LPEPS-1 was 42.6%, which was lower than that of Vc at 4.0 mg/mL by half, indicating that LPEPS-1 had moderate scavenging activity on DPPH. The scavenging activity was similar to Wang et al. [17], who reported that the DPPH radical scavenging activity of r-EPS1 and r-EPS2 isolated from *L. plantarum* 70810 was 27.98% and 48.43%, respectively. The DPPH free radical scavenging rate was mainly affected by the monosaccharide composition. In addition, it has been reported that the high molecular weight of polysaccharide may affect its solubility in water–ethanol systems, thus affecting its scavenging ability [38].

#### 3.7.2. Superoxide Anion Free Radical Scavenging Activity of EPS

Superoxide anion is a precursor of active free radicals, and it is a relatively weak oxidant [45]. When it interacts with singlet oxygen or hydroxyl radicals, it can generate strong reactive oxidative species and cause oxidative damage and various diseases [46]. Therefore, the superoxide radical scavenging activity of LPEPS-1 was evaluated. As shown in Figure 5b, the scavenging ability of LPEPS-1 against the superoxide anion showed low scavenging ability at the low concentration range (0.125 mg/mL to 0.5 mg/mL). Its scavenging ability was less than 30%. As the concentration of polysaccharides increased, the scavenging activity gradually improved. When the test concentration was up to 4.0 mg/mL, the highest scavenging activity was 40.13%. Compared with Vc, the scavenging activity of LPEPS-1 was relatively low. However, LPEPS still exhibited similar antioxidant activity to what has been reported in the literature [22].

#### 3.7.3. Metal Ion-Chelating Activities

Metal ions, such as ferrous ions and copper ions, are powerful oxidants. It is reported that they can induce the production of free radicals and reactive oxygen species, which can lead to lipid peroxidation and DNA damage [47]. The Fe^2+^-chelating activity of LPEPS-1 was evaluated (Figure 5c). For chelating activity on ferrous ion, LPEPS-1 showed excellent chelating ability. As the concentration of polysaccharides increased, the chelating ability remarkably increased. The chelating ability of LPEPS-1 on Fe^2+^ at 4.0 mg/mL was up to 62.8%, which was close to the result obtained for EDTA-2Na. Additionally, previous studies have shown that compounds containing -OH, C=O, -SH, -O-, C=O, and other functional groups have good metal chelation [48]. These results indicate that the -OH, -O-, and C=O functional groups in LPEPS-1 played a role in metal chelation.

The antioxidant property of polysaccharide has been a hot topic. Recent studies have reported that the antioxidant property of polysaccharide was related to many factors, such as molecular weight, monosaccharide compositions, molar ratio of monosaccharides, glycoside bonds, and others [49,50]. Lo et al. [51] reported that the composition of monosaccharides affects the antioxidant capacity of EPS and that its monosaccharides containing rhamnose and galactose have good antioxidant capacity. In our studies, LPEPS-1 contained a galactose composition, so it exhibited outstanding antioxidant activity. Although there are many research studies on the antioxidant activity of polysaccharides, there are few studies on the antioxidant mechanism, which needs further exploration.

## 4. Conclusions

In this study, one purified EPS fraction (LPEPS-1) was extracted from soybean whey fermented by *L. plantarum* 70810, and the structural characteristics and antioxidant activity of LPEPS-1 were investigated. Results show that LPEPS-1 consisted of mannose, glucose, and galactose, with molar ratios of 1.49:1.67:1.00. Particle size analysis showed that LPEPS-1 had a small particle size and that it was negatively charged. Finally, we also evaluated its antioxidant activity in vitro. LPEPS-1 exhibited better antioxidant bioactivity against the DPPH radical scavenging activity, superoxide anion radical scavenging activity, and chelating activity on ferrous ion. On the basis of these results, *L. plantarum* 70810 can be used with soybean whey to produce polysaccharide, and the EPS has a variety of antioxidant activities, providing evidence of its further research value.

## Figures and Tables

**Figure 1 foods-10-02780-f001:**
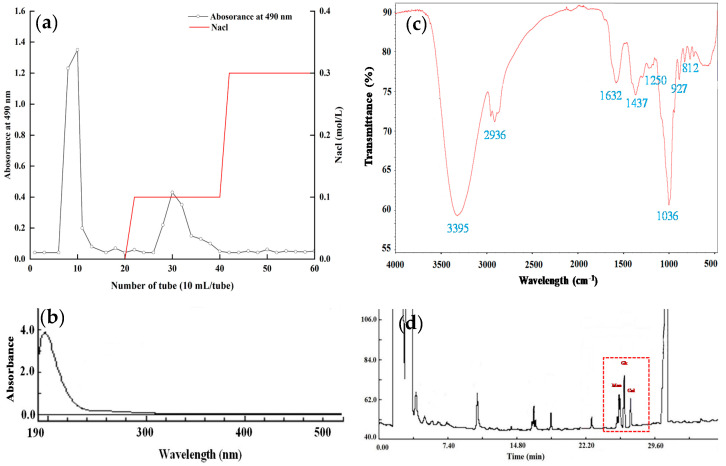
Stepwise elution curve of EPS produced by *L. plantarum* 70810 on DEAE cellulose-52 chromatography column (**a**); UV spectrum (**b**), FT-IR spectrum (**c**), GC chromatogram of monosaccharide composition of LPEPS-1 (**d**).

**Figure 2 foods-10-02780-f002:**
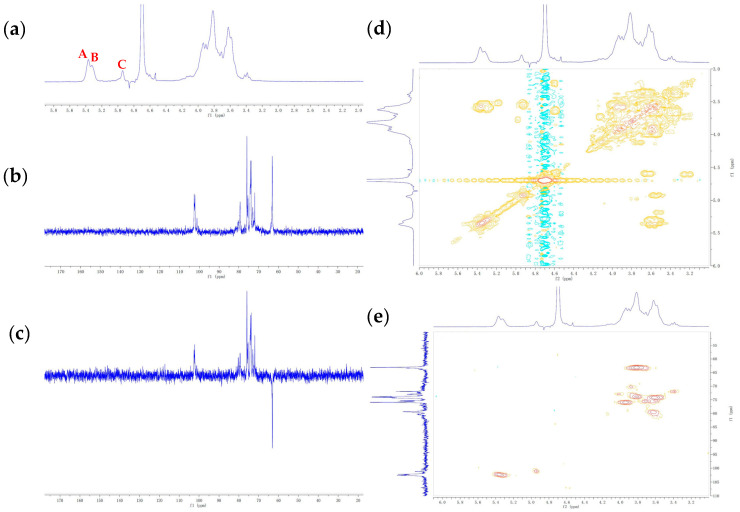
NMR spectra of LPEPS-1. ^1^H NMR (**a**), ^13^C NMR (**b**), DEPT-135 ^13^C NMR (**c**), ^1^H-^1^H COSY (**d**), and ^1^H-^13^C HSQC of LPEPS-1 (**e**).

**Figure 3 foods-10-02780-f003:**
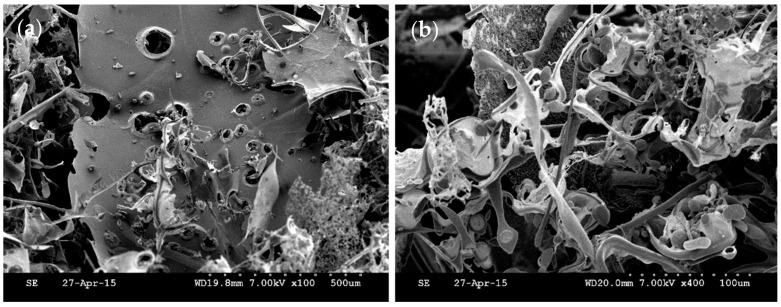
SEM images of LPEPS-1 from *L. plantarum* 70810. ×100 LPEPS-1 (**a**), ×400 LPEPS-1 (**b**).

**Figure 4 foods-10-02780-f004:**
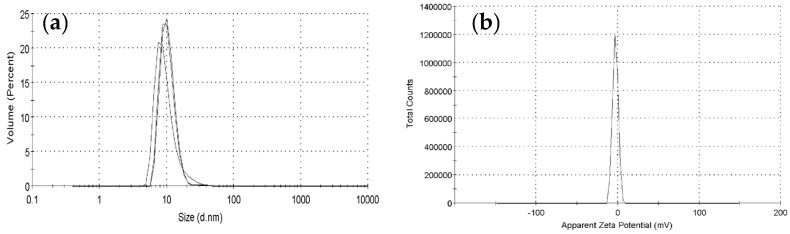
Particle size distribution and apparent zeta potential of LPEPS-1. Particle size distribution (**a**), apparent zeta potential (**b**).

**Figure 5 foods-10-02780-f005:**
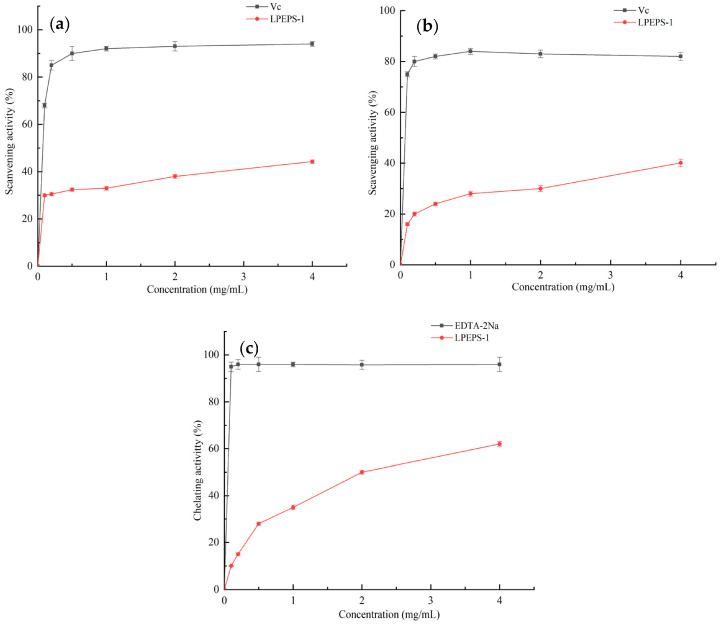
In vitro antioxidant activity of LPEPS-1. Scavenging activity on DPPH radical (**a**), scavenging activity on superoxide radical (**b**), chelating activity on metal ion (**c**), Vc and EDTA-2Na as positive control. Data presented as means ± SD triplicates.

**Table 1 foods-10-02780-t001:** Chemical shifts (ppm) of ^1^H and ^13^C signals for the LPEPS-1 recorded in D_2_O at 313 K.

Residues	Sugar Linkages	H-1/C-1	H-2/C-2	H-3/C-3	H-4/C-4	H-5/C-5	H-6/C-6
A	→4)-α-Glc*p* (1→	5.37 102.19	3.58 73.88	3.93 75.13	3.67 79.52	3.78 73.88	3.82 63.11
B	→2)-α-Man*p* (1→	5.32 102.57	3.60 79.24	3.90 70.88	3.69 71.94	3.89 69.53	3.74 62.40
C	→3)-α-Gla*p* (1→	4.94 101.10	3.58 74.38	3.97 76.10	4.00 72.37	3.72 75.50	3.80 63.16

## Data Availability

Not applicable.

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
