# Peer review of "Structural Characterization and Antioxidant Activity of Exopolysaccharide from Soybean Whey Fermented by Lacticaseibacillus plantarum 70810"

_foods, 2021, doi:10.3390/foods10112780_

Round 1
Reviewer 1 Report
The subject of the work is interesting. It concerns functional food and searching for ways of waste management in the food industry.
- 38-40: Unclear wording - clarify or better structure the sentence in English;
- 147-148: The method for determining the activity against DPPH radicals should be better described - why were they made to different final concentrations? Why was ethanol used, and not the commonly used methanol in this method, which gives better results in this assay? Why is the antiradical activity given in% - in such units the results become difficult to compare?
- 154: This method should be described in more detail - how was the radical generation generated?
- 163: The described method for testing the ability to chelate iron ions is based on a standard curve. The procedure should be described. How do you directly calculate the chelating capacity from the formula? In the method, we measure the amount of non-chelated iron. The result, without conversion from the standard curve, cannot be substituted for such a formula. This needs improvement.
- 288: “Besides the purity and color of the polysaccharide also can influence the results” - this method uses a blank and a control to eliminate this - this should be clarified. Such an impact should not take place.
You should adapt the literature list to the journal's requirements - this applies to commas, periods and italics.
Author Response
Review1#:
The subject of the work is interesting. It concerns functional food and searching for ways of waste management in the food industry.
1. 38-40: Unclear wording - clarify or better structure the sentence in English;
Response: Thank you for your valuable comments. In the revised manuscript, we have revised the unclear sentence (Line 38-40).
2. 147-148: The method for determining the activity against DPPH radicals should be better described - why were they made to different final concentrations? Why was ethanol used, and not the commonly used methanol in this method, which gives better results in this assay? Why is the antiradical activity given in% - in such units the results become difficult to compare?
Response: Thank you for your valuable comments. In the revised manuscript, the method for determining the activity against DPPH radicals has been better described (Line 145-148). In addition, the “final concentrations” means “Solid polysaccharides were added deionized water to prepare a certain concentration of polysaccharide solution”. The word “final” maybe inaccurate, so it has been deleted. Since DPPH is relatively stable in organic reagents, ethanol is used to dissolve DDPH. In this method, some people use methanol, but according to the literature, ethanol uses relatively more (Yin et al., 2021; Shan et al., 2012; Qiao et al., 2009). The results give in %, which is currently the most used in the literature, and it is calculated relative to the blank sample. In addition, we also used VC as a positive control. So the results can be compared with each other.
3. 154: This method should be described in more detail - how was the radical generation generated?
Response: Thank you for your valuable comments. In revised manuscript, the method of scavenging activity of superoxide anion has been described in more detail. In addition, the generation of superoxide anion was also added. The reaction principle is that PMS reacts with NADH to produce superoxide anions. The generated superoxide anion can react color with NBT. If the polysaccharide solution has the ability to clear superoxide anions, its reaction color will become lighter (Line156-159).
4. 163: The described method for testing the ability to chelate iron ions is based on a standard curve. The procedure should be described. How do you directly calculate the chelating capacity from the formula? In the method, we measure the amount of non-chelated iron. The result, without conversion from the standard curve, cannot be substituted for such a formula. This needs improvement.
Response: Thank you for your valuable comments. The method you mentioned is different from the method we refer to. Our method refers to a lot of literatures (Liu et al., 2007; Qiao et al., 2009; Li et al., 2018), they do not need standard curve and standard products, and the literatures are calculated according to this formula. Fecl2 can chelate with Felozine to form Felozine-Fe2+, however, when polysaccharide solution is added, the polysaccharide can chelate with Fe2+, thereby inhibiting Felozine-Fe2+ formation. Therefore, the chelating rate was calculated by measuring the absorbance change before and after adding the polysaccharide by an ultraviolet-visible spectrophotometer at 562 nm.
5. 288: “Besides,the purity and color of the polysaccharide also can influence the results” - this method uses a blank and a control to eliminate this - this should be clarified. Such an impact should not take place.
Response: Thank you for your valuable comments. In the experimental method, the purity and color of the polysaccharide have been taken into consideration, so this discussion is not appropriate. In the revised manuscript, we deleted this discussion.

Reviewer 2 Report
The article is well written and presents relevant results. But, there are several mistakes as described in the attached file.
- the references mainly in the introduction section should be updated
- English has to be checked.
- in the material and method section, the statistics are not clearly presented, and also in the results section I could not find any clear indication of how it was performed
- the result discussion has to be detailed
- please cite references according to journal guideline
Otherwise, the article is interesting.

Author Response
Here are our point-by-point responses to the reviewers’ comments/ questions:
Review2#:
The article is well written and presents relevant results. But, there are several mistakes as described in the attached file.
1. the references mainly in the introduction section should be updated.
Response: Thank you for your valuable comments. In the revised manuscript, we have readjusted the references in the introduction
2. English has to be checked.
Response: Thank you for your nice caution. Some writing and grammar mistakes in lines of 26, 28, 30, 44, 45, 57, 183, 193, 202, 220, 221, 260, 261, 266 and 319-321 have been revised in the corresponding position of the revised manuscript. In addition, the full manuscript was carefully checked and other writing and grammar mistakes were also corrected with green font.
3. in the material and method section, the statistics are not clearly presented, and also in the results section I could not find any clear indication of how it was performed.
Response: Thank you for your valuable comments. In the revised manuscript, the statistical analysis has been re-described (Line 174-176).
4. the result discussion has to be detailed
Response: Thank you for your valuable comments. We have expanded the discussions to the revised manuscript (Line 277-280, Line 282-284, Line 298-304, Line 306-308, Line313-315).
5. please cite references according to journal guideline
Response: Thank you for your valuable comments. The reference format has been cited in accordance with the guideline.
6. Otherwise, the article is interesting.
Response: Thank you for your interest in our manuscript.

Round 2
Reviewer 1 Report
the corrections have been made.
Author Response
Thank you so much.
Reviewer 2 Report
Dear authors,
The article was partially improved, but I still can`t find an update of the literature in the introduction section. There are several sentences where a reference is missing, or the authors state that nowadays and introduce references older than 9 years. Please revise the english also, and the quality of figures have to be improved.
In the attached file there are some specific comments, the authors should revise the whole manuscript, and also correct is accordingly.

Author Response
Here are our responses to the reviewers’ 2 comments:
Review2#:
- Thank you for your comments. In the second revised manuscript, the references in the introduction have been updated to the latest references as much as possible. In addition, the lack of references have also been added in the introduction. (E.g. references [5] and [10]). The updated and lack of references have been marked in green in the reference section.
- Line 123, 'Gremany' has been revised 'Germany'
- The quality of Figure 5 has been improved.
